# Quinolone Resistance Genes and Their Contribution to Resistance in *Vibrio cholerae* Serogroup O139

**DOI:** 10.3390/antibiotics12020416

**Published:** 2023-02-20

**Authors:** Yan-Yan Zhou, Li-Yan Ma, Li Yu, Xin Lu, Wei-Li Liang, Biao Kan, Jian-Rong Su

**Affiliations:** 1Department of Clinical Laboratory Medicine, Beijing Friendship Hospital, Capital Medical University, Beijing 100050, China; 2Beijing Municipal Center for Disease Prevention and Control, Beijing 100013, China; 3Chinese Center for Disease Control and Prevention, State Key Laboratory for Infectious Disease Prevention and Control, Department of Diarrheal Diseases, National Institute for Communicable Disease Control and Prevention, Beijing 102206, China

**Keywords:** *Vibrio cholerae* O139, *qnrVC*, quinolone resistance, plasmid

## Abstract

Background. Quinolones are commonly used for reducing the duration of diarrhea, infection severity, and limiting further transmission of disease related to *Vibrio cholerae*, but *V. cholerae* susceptibility to quinolone decreases over time. In addition to mutations in the quinolone-resistance determining regions (QRDRs), the presence of *qnr* and other acquired genes also contributes to quinolone resistance. Results. We determined the prevalence of quinolone resistance related genes among *V. cholerae* O139 strains isolated in China. We determined that eight strains carried *qnrVC*, which encodes a pentapeptide repeat protein of the Qnr subfamily, the members of which protect topoisomerases from quinolone action. Four *qnrVC* alleles were detected: *qnrVC1*, *qnrVC5*, *qnrVC12*, and *qnrVC9*. However, the strains carrying *qnrVC1*, *qnrVC5*, and *qnrVC12* were ciprofloxacin (CIP)-sensitive. Contrastingly, the strain carrying *qnrVC9* demonstrated high CIP resistance. *qnrVC9* was carried by a small plasmid, which was conjugative and contributed to the high CIP resistance to the receptor *V. cholerae* strain. The same plasmid was also detected in *V. vulnificus*. The *qnrVC1*, *qnrVC5*, and *qnrVC12* were cloned into expression plasmids and conferred CIP resistance on the host *V. cholerae* O139 strain. Conclusions. Our results revealed the contribution of quinolone resistance mediated by the *qnrVC9* carried on the small plasmid and its active horizontal transfer among *Vibrio* species. The results also suggested the different effects of *qnrVC* alleles in different *V. cholerae* strains, which is possibly due to differences in sequences of *qnrVC* alleles and even the genetic characteristics of the host strains.

## 1. Introduction

Cholera is an acute gastrointestinal tract disease caused by *Vibrio cholerae*, of which two serogroups (O1 and O139) demonstrate epidemic potential. To date, seven major cholera pandemics have occurred since the 19th century, the most recent of which originated in Indonesia in the 1960s and is still ongoing [1]. Major cholera epidemics that have occurred periodically in recent years were those in Zimbabwe (2008), Haiti (2010), Sierra Leone (2012), Mexico (2013), and South Sudan and Ghana (2014) [1]. Despite the availability of effective prevention and treatment methods, cholera persistence remains a major public health problem worldwide mainly due to socioeconomic factors such as unsatisfactory hygienic conditions. The World Health Organization (WHO) estimated that 3–5 million cholera cases occur globally every year, with 100,000–120,000 deaths [2]. Since the first 1993 outbreak caused by the *V. cholerae* O139 strains in Xinjiang, China [3], limited foodborne outbreaks associated with food poisoning and sporadic cases were frequently reported and expanded in various regions, especially southeastern China [4]. The organisms resided and survived within environmental reservoirs during inter-epidemic periods and the threat of a cholera epidemic is ever-present.

Antibiotic therapy reduces the duration of diarrhea and infection severity, and limits further transmission of disease related to *V. cholerae* [5]. Currently, azithromycin and ciprofloxacin (CIP) are commonly used for treating patients with cholera [6]. CIP belongs to the quinolones, a family of broad-spectrum, systemic antibacterial agents that act by inhibiting DNA gyrase (*gyrA* and *gyrB*) and DNA topoisomerase IV (*parC* and *parE*), which are required for bacterial mRNA synthesis and DNA replication [7]. Due to excessive antibiotic use and misuse in humans, agriculture and aquaculture systems, antibiotic resistance emerged and evolved in bacteria, including *Vibrio* species, during the past few decades [8]. Bacteria often combine more than one mechanism to increase drug resistance acquisition efficacy. In addition to mutations in the quinolone-resistance determining regions (QRDRs) and overexpression of efflux pumps that reduce intracellular concentrations of the drug [9], Qnr proteins and the inactivation of drug by quinolone-modifying enzyme (AAC (6′) Ib-Cr) [10] also contribute to quinolone resistance.

In a previous survey, we examined the *V. cholerae* O139 strains recovered in 1993–2009 in China and detected strains that were increasingly resistant to nalidixic acid (NAL), which is the first member of the quinolones, and decreasingly sensitive to CIP [4]. Subsequent analysis revealed that the accumulation of mutations in the DNA gyrase and topoisomerase IV genes contributed to fluoroquinolone resistance [11]. However, it is unclear whether quinolone resistance related genes contribute to quinolone resistance in *V. cholerae*. Therefore, we investigated the prevalence of quinolone resistance related genes in *V. cholerae* serogroup O139 strains and estimated whether they could mediate quinolone resistance to their host strains.

## 2. Results

### 2.1. Prevalence of Quinolone Resistance Related Genes

In a previous study, we conducted a comprehensive investigation of the antibiotic resistance of *V. cholerae* O139 strains isolated in China from 1993 to 2009. The multidrug resistance in O139 isolates increased suddenly and became common after 1998. Different resistance profiles were observed in the *V. cholerae* O139 isolates from different years, while *V. cholerae* O1 strains isolated in the same period were much less resistant to these antibiotics and no obvious multidrug resistance patterns were detected. We further found decreased susceptibility was exhibited to CIP in *V. cholera* O139 strains [4]. Between 2001 and 2009, the median MIC of CIP was 0.5 μg/mL, without a change over this interval. These values were 16.7-fold and 4.2-fold higher than the median MICs of CIP for isolates of V. cholerae in 1993–1997 (0.03 μg/mL) and 1998–2000 (0.12 μg/mL), respectively. The seven strains resistant to CIP, which carried these accumulated mutations in QRDRs, were isolated in 1998, 2001 and 2003, respectively [11]. In 2006, there was one isolated strain named VC1699 possessing no mutation in QRDR of *parE* but resistant to CIP.

To determine whether quinolone resistance related genes contributed to quinolone resistance in *V. cholera,* we performed PCR screening for the prevalence of these genes in *V. cholerae* O139 strains. *qnrVC* genes were detected from eight *V. cholerae* O139 strains: VC422, VC1435, VC707, VC454, VC319, VC515, VC1692, and VC1699. Furthermore, QRDR genes were also detected from these strains and sequenced to identify existing mutations. Theses strains contained a series of mutations such as the meaningful mutations GyrA S83I and ParC S85L and nonsense mutations such as GyrA A171S.

MEGA 7.0 alignment of the *qnrVC* PCR amplicon revealed a series of *qnrVC* alleles. The *qnrVC* allele amplified from VC422, VC515, and VC707 was identical to *qnrVC1*; that from VC454, VC319, and VC1692 was identical to *qnrVC5*; and that from VC1699 was identical to *qnrVC9*. Lastly, the *qnrVC* allele from VC1435 differed from *qnrVC4* by two single amino acids (Met18Leu and Gly167Glu) and was designated *qnrVC12* (GenBank accession no. OQ442957). The QnrVC12 protein sequence was found to have 99% identity with QnrVC4. Table 1 lists the main characteristics of these eight strains. Phylogenetic analysis showed the QnrVC12, QnrVC4, QnrVC7, QnrVC9, and qnrVC5 belonged to the same group, whereas QnrVC10, QnrVC1, QnrVC3, QnrVC6, and QnrVC8 formed another group (Figure 1).

### 2.2. Characterization of Plasmids Containing qnrVC9

Among the eight strains containing *qnrVC* genes, we extracted a plasmid termed pVC1699 from VC1699, which was isolated from a patient in Jiangxi in 2006. Sequencing of pVC1699 determined that it was 7081-bp long with a GC content of 42.85%. The plasmid sequence was submitted to GenBank under the accession number OP821998. The ORF search revealed that pVC1699 carried qnrVC9 genes related to quinolone resistance. pVC1699 also carried genes for antitoxin (*parD1*), plasmid stabilization system protein (*parE*), integrase 1 (*int1*), and dihydrofolate reductase type 1 (*dhfrI*). BLAST revealed that pVC1699 had low identity with plasmids harbored *qnrVC5,* including pBD146 from *V. fluvialis* (GenBank accession no. EU574928.1), plasmids v110 from *V. parahaemolyticus* (GenBank accession no. KC540630.1), and pVN84 from *V. cholerae* O1 (GenBank accession no. AB200915.1) (Figure 2). However, pVC1699 had high identity to 99% with a plasmid termed 307 from *V. vulnificus* (GenBank accession no. MZ325519) (Figure 2).

### 2.3. Effect of qnrVC on Quinolone Resistance

Except VC1699, the *qnrVC*-containing strains were CIP-sensitive (Table 1). A series of plasmids were constructed to determine the effect of different *qnrVC* alleles on quinolone resistance (Table 2). First, when plasmid pVC1699 was translated into VC401, the CIP MIC of the constructed strain increased from 0.375 µg/mL to 4/6 µg/mL, a 10.7/16.0-fold increase. There was a slight difference among the CIP MIC for the same VC401 strains harboring pBAD24-*qnrVC1*, -*qnrVC5*, -*qnrVC12*, and -*qnrVC9*. Expressing *qnr9* genes from VC1699 through vector pBAD24 in VC401, the CIP MIC was 3/4 µg/mL, and inducing the expression of the *qnrVC1*, *qnrVC5*, and *qnrVC12* genes from the quinolone-sensitive strains through pBAD24 increased the CIP MIC to 1.5–2 μg/mL. Furthermore, expressing *qnrVC9* genes through vector pBAD24 in N16961, a reference *V. cholera* O1 strain without QRDR mutations, the CIP MIC was increased from 0.015 to 0.125 μg/mL, an 8.3-fold increase. Moreover, expressing *qnrVC9* genes through vector pBAD24 in VC1891, a *V. cholera* O139 stain harboring nonsense A171S mutation in QRDR, the CIP MIC was increased from 0.03 to 0.125/0.25 μg/mL, a 4.2/8.3-fold increase.

As *qnr* expression required induction by arabinose, and the CIP MICs of stains containing pBAD24-*qnrVC9* were lower than that of stains harboring plasmid pVC1699, the mRNA expression level of *qnrVC9* was also analyzed by RT-PCR. As shown in Figure 3, the mRNA expression level of *qnrVC9* in pBAD24 in N16961, VC1891, and VC401 were higher than that in pVC1699 in VC401.

## 3. Discussion

Qnr, encoding by *qnr* gene, was a pentapeptide repeat protein of 218 amino acids which protect DNA gyrase from quinolone action [12]. *qnr* gene was first discovered in a plasmid from *Klebsiella pneumoniae* in 1998 [13], and increasingly being reported worldwide from bacteria isolated from both clinical and aquatic environments [14]. To date, several *qnr* gene families have been identified, mainly containing *qnrA*, *qnrB*, *qnrC*, *qnrD*, *qnrE*, *qnrS*, and *qnrVC* [14,15,16,17,18]. These *qnr* genes generally differ in sequence by 35% or more from each other. Allelic variants differing by 10% or less have also been described in almost every *qnr* family [9]. Since the *qnrVC1* gene was first described in a class 1 integron from a *V. cholerae* O1 strain recovered in 1998 [15], twelve *qnrVC* alleles (*qnrVC1–12*) have been reported, mainly in *Vibrionaceae* to date [16,19,20,21,22,23], as well as in *Enterobacterales* in Brazilian coastal waters [24]. *qnrB, qnrC, qnrD*, and *qnrA* were harbored by 20-35%, *aac(6’)-Ib-cr* was harbored by 15% of the CIP unsusceptible *P. aeruginosa* isolates [25]. The predominant resistant genes were *aac(6’)-Ib-cr* (48.9%) and *qnrD* (25.6%) in fluoroquinolone resistant *E. coli* isolates [26]. In our survey, *qnrVC* genes were more prevalent in *V. cholerae* as compared to other quinolone resistance related genes. *qnrVC* was harbored by 2.4% (8/340) of *V. cholera* stains, and *aac(6’)-Ib-cr* was not found in them. In our study, *qnrVC1* was discovered in 2002, then *qnrVC12* in 2003, *qnrVC5* in 2004, and the *qnrVC9* was discovered in 2006, demonstrating the undergone successive mutations over time.

The most common mechanism of high-level quinolone resistance was due to accumulation of mutations within QRDRs for both Gram-negative and Gram-positive organisms [27]. Usually, bacteria carrying *qnr* genes only presented low-level quinolone resistance by blocking CIP inhibition of topoisomerase IV or DNA gyrase [28]. For example, the CIP MIC of *V. cholerae* O1 carrying *qnrVC1* was 0.25 µg/mL [15], the CIP MIC of *V. cholerae* O1 strain N16961 carrying *qnrVC9* was 0.125 µg/mL in our study. Furthermore, Qnr enhanced the frequency of spontaneous mutations in DNA gyrase and topoisomerase IV genes when facing antibiotic stress [13]. The *qnr*-bearing strains contained a series of spontaneous mutations in the QRDR genes, but only VC1699 contained mutations S83I in GyrA and S85L in ParC, which typically caused the CIP MIC to increase up to 0.25–2 µg/mL [11]. When combined with QRDR gene mutations and with an efflux pump, a high-level fluoroquinolone resistance phenotype easily emerged and resulted in treatment failure [23]. The CIP MIC of *V. cholerae* O1 carrying *qnrVC3* and QRDR gene mutations was 0.5–0.75 µg/mL [29]. The CIP MIC of *V. fluvialis* carrying *qnrVC3* and QRDR gene mutations was 2.5 µg/mL [12]. In our study, *qnrVC9* resided on plasmid pVC1699 combined with QRDR gene mutations could caused higher-level CIP resistance. It was also reported *qnrB, qnrS combined with* QRDR gene mutations could cause higher-level CIP resistance in Enterobacteriaceae [30].

The drug resistance genes often resided on mobile genetic elements for easy dissemination of drug resistance to other organisms. The mobile genetic elements, especially plasmids, played a major role in the dissemination of antimicrobial resistance [31]. *qnrVC* alleles were usually associated with class 1 integron, SXT or plasmid [22,23]. Interestingly, *qnr* genes are found in a variety of Gram-positive organisms but are chromosomal and not plasmid-mediated [9]. *qnrVC9* was first reported residing in the chromosome of a *V. cholera* isolate obtained from a pork sample in China [19] and we determined that Int1 captured *qnrVC9* and inserted it in a plasmid. The *qnrVC9* on the plasmid rather than on the chromosome would transfer more easily to other bacteria, indicating that *qnrVC9* has the same mobility and dispersion through different hosts as other plasmid-located *qnrVC* alleles.

Due to the extensive usage of antimicrobials, the presence and accumulation of quinolones in waters has been widely reported [32]. The aquatic environment played a highly important role in the transfer and maintenance of bacterial genes encoding for antibiotic resistance [33,34] due to the antimicrobials selective pressure. *qnr* genes carried by Gram-negative bacteria were extensively detected in bacteria isolated from aquatic environments [14]. pVC1699, from a *V. cholera* O139 strain isolated in 2006 in Jiangxi Province, was entirely identical to plasmid 307 isolated from *V. vulnificus*, whose sequence was submitted in 2021 from Shanghai, China. It appeared that pVC1699 could transmit between *V. vulnificus* and *V. cholerae* O139, and this plasmid was maintained in *Vibrios* for years. It was also reported other qnr-harboring plasmids could transfer within different hosts under antibiotic induced stress [31]. Therefore, more attention should be paid to this highly efficient plasmid pVC1699 in the capture and dissemination of *qnrVC9* among *Vibrio* species.

Except for strain VC1699, the quinolone-sensitive strains also carried *qnrVC* genes. In these strains, *qnrVC* might increase the frequency of spontaneous mutations in DNA gyrase and topoisomerase IV genes under antibiotic stress [13]. In our study, the CIP MIC increased by varying degrees when *qnrVC1*, *qnrVC5*, and *qnrVC12* gene expression was induced through pBAD24, which suggested that *qnrVC1*, *qnrVC5*, and *qnrVC12* also can contribute to low-level quinolone resistance in *V. cholera*. This phenomenon was also described in other study, pCN60-*qnrVC1* vector elevated the MIC of CIP from 0.003 to 0.25 µg/mL in *E. coli,* but seemed to have no effect in *P. aeruginosa* [35]. This different effect was possibly due to the different efficiency of the promoter or the protein in different strains [35].

The three-dimensional structure of Qnr proteins has been determined by X-ray crystallography [36]; essential amino acids were found in the pentapeptide repeat module and in the larger of two loops, where deletion of only a single amino acid compromises activity [37]. The amino acid change in QnrVC7 also could increase or decrease the CIP MIC: the CIP MIC of *E. coli* harboring pCR2.1-*qnrVC7* was 0.06 µg/mL, *E. coli* harboring pCR2.1-*qnrVC7*-T152A was 0.25 µg/mL, and *E. coli* harboring pCR2.1-*qnrVC7* -A82T was 0.015 µg/mL [38]. It was supposed that the difference in sequences of *qnrVC* alleles resulted in the CIP MIC of VC401 strains carrying pBAD24-*qnrVC1, -5, -10, -9* were slightly different from each other. In other reports, different *qnrVC* alleles also showed different effect on susceptibility to CIP. For example, *E. coli* harboring pET-*qnrVC5* vector had a more than 64.2-fold increase CIP MIC from <0.000243 to 0.0156 µg/mL [12]; *E. coli* harboring pGEMT-qnrVC1 vector had a 41.7-fold increase CIP MIC from 0.003 to 0.125 µg/mL [35]; *E. coli* harboring pCR2.1–*qnrVC6* vector had a more than 5-fold increase CIP MIC from <0.05 to 0.25 µg/mL [16]. However, despite the *qnrVC* gene over-expression in VC401/pBAD24-*qnrVC9*, its CIP MIC was still slightly lower than that of VC401/pVC1699. This might reflect a complex interplay of many known and yet unknown genetic factors. This similar phenomenon was also reported in other study; *qnrVC1* seemed to have no effect when harboring by another *P. aeruginosa*. The CIP MIC of *P. aeruginosa* (pCN60*qnrVC1*) was only 0.19 µg/mL, compared with >32 µg/mL for the native strain *P. aeruginosa* Pa25 [35].

## 4. Materials and Methods

### 4.1. Bacterial Strains

A total of 340 *V. cholerae* O139 strains recovered from patients or the environment in 1993–2009 were randomly selected from the *V. cholerae* strain bank at the Chinese Center for Disease Control and Prevention (Beijing, China) and were used in our previous study [4], including 290 toxigenic (*ctxAB* gene positive) strains and 50 non-toxingenic (*ctxAB* genes negative) strains. These strains, including 237 isolated from patients; 8 isolated from healthy carrier; 13 isolated from fish, bullfrog, or soft-shelled turtle; 45 from surface swabs of patient-related places, or river or sewage water samples; and 37 lacking the related information from the isolated province. A typical wild-type *V. cholerae* O139 strain, VC401, harbored *ctxAB* gene was chosen as the recipient strain for electroporation experiments. VC401 had a CIP minimum inhibitory concentration (MIC) of 0.375 µg/mL due to the prevalent double mutations of S83I in GyrA and S85L in parC.

### 4.2. Antibiotic Susceptibility Testing

Antibiotic susceptibility tests were performed using the agar dilution method according to the current guidelines of the Clinical and Laboratory Standards Institute (CLSI). The precise CIP MIC of the cloning strains was determined with intensive serial dilutions: 0.007, 0.015, 0.03, 0.06, 0.125, 0.25, 0.375, 0.5, 1, 1.5, 2, 3, 4, 6, 8 µg/mL. The control for the antibiotic susceptibility tests was *Escherichia coli* ATCC 25922.

### 4.3. PCR and DNA Sequencing

The quinolone resistance related genes (*qnr* [13], *aac(6′)-Ib-cr* [10], and *qepA*, which encoded an efflux pump [39]) were amplified by PCR using previously described primers (Table 3) [29,32]. The PCR primers for detecting *ct, int1, sxt, gyrA, gyrB, parC,* and *parE* were described in our previous publication [4]. Furthermore, we amplified and sequenced the open reading frame (ORF) of *qnr* genes from target strains using the primers qnrF/qnrR or 2qnrF/2qnrR (Table 3). Total chromosomal DNA from *V. cholerae* O139 was prepared with a DNA purification kit (Tiangen Biotech, Beijing, China). PCR amplification was performed with 2× Taq PCR MasterMix (Tiangen Biotech, China). PCR amplicons were sequenced commercially (Shanghai Sangon Biotech, China) and aligned using MEGA 7.0 software. A phylogenetic tree was constructed using Neighbor-Joining method in MEGA 7.0 software after multiple sequence alignments via CLUSTAL 2.1.

### 4.4. Plasmid Extraction, Electroporation and Sequencing

Plasmids from all *qnr*-positive *V. cholerae* strains were extracted using a QIAprep Spin Miniprep Kit (Qiagen, Valencia, CA, USA). The plasmids were sequenced by primer walking and analyzed by National Center for Biotechnology Information (NCBI) BLAST. The possible ORFs were predicted using ORFfinder (https://www.ncbi.nlm.nih.gov/orffinder/, accessed on 1 November 2022). Plasmid transformation into *V. cholerae* was performed by electroporation with selection on Luria–Bertani (LB) agar containing CIP (1 μg/mL). PCR screening was performed for the QRDRs mutations of the VC401/pVC1699 strains to rule out the possibility of any point mutations.

### 4.5. Cloning and Expression of qnrVC

The *qnrVC* genes from *V. cholerae* wild-type strains were cloned into expression vector pBAD24 using the primers *qnrEcor*I and *qnrHind*III for VC454, VC1435, and VC1699 and primers *2qnrF*/*2qnrR* for VC422 and VC707, and then transformed into *E. coli* TOP10. The constructed plasmids, which were pBAD24-*qnrVC1*, -*qnrVC5*, -*qnrVC12*, and *-qnrVC9*, were validated by sequencing. These plasmids were transformed into *V. cholerae* VC401 by electroporation with selection on LB agar containing ampicillin (100 μg/mL). PCR screening was performed for the QRDRs mutations of the VC401/pBAD24-*qnrVC* strains to rule out the possibility of any point mutations. As *qnr* expression required induction by arabinose, VC401 harboring pBAD24-*qnrVC* was first cultured at 37 °C on LB agar containing 0.1% (*w*/*v*) arabinose, and then the CIP MICs of the donor and recipient strains were compared by antibiotic resistance testing using Mueller–Hinton agar also containing 0.1% (*w*/*v*) arabinose. All experiments were replicated three times.

RT-PCR was mainly carried out to confirm the expression of *qnrVC9* gene in the native isolate VC1699 and the series of transformants. Total RNA was isolated with the RNAprep Pure Cell/Bacteria Kit (Tiangen Biotech, China). The RNA preparations were subsequently treated with DNaseI (TaKaRa, Kusatsu, Japan) to remove the genomic DNA contamination. Reverse transcription was carried on by PrimeScript RT reagent kit (TaKaRa, Japan). Analysis of gene expression levels using TB Green Premix Ex Taq II (TaKaRa, Japan) was carried out on Applied Biosystems 7300 Real-Time PCR System (Thermo Scientific, Waltham, MA, USA) following the protocol recommended by the manufacturer. The *qnrF2*/*qnrR2* primers were used for qnr RT-PCR, and *recA* was used as an internal control with primers *recAF*/*recAR*. The relative expression values (R) were calculated using the equation R = 2(-Delta Delta C(T)) (where CT is the fractional threshold cycle) [40].

## 5. Conclusions

In summary, we determined the prevalence of quinolone resistance related genes among *V. cholerae* O139 strains isolated in China. Four *qnrVC* alleles were detected, but only the strain carrying *qnrVC9* had high quinolone resistance. *qnrVC9* was carried by a small plasmid, which was conjugative and contributed high CIP resistance to the receptor *V. cholerae* strain. The same plasmid was also detected in *V. vulnificus.* Our findings expand our knowledge on the transmission of quinolone resistance among *Vibrios*.

## Figures and Tables

**Figure 1 antibiotics-12-00416-f001:**
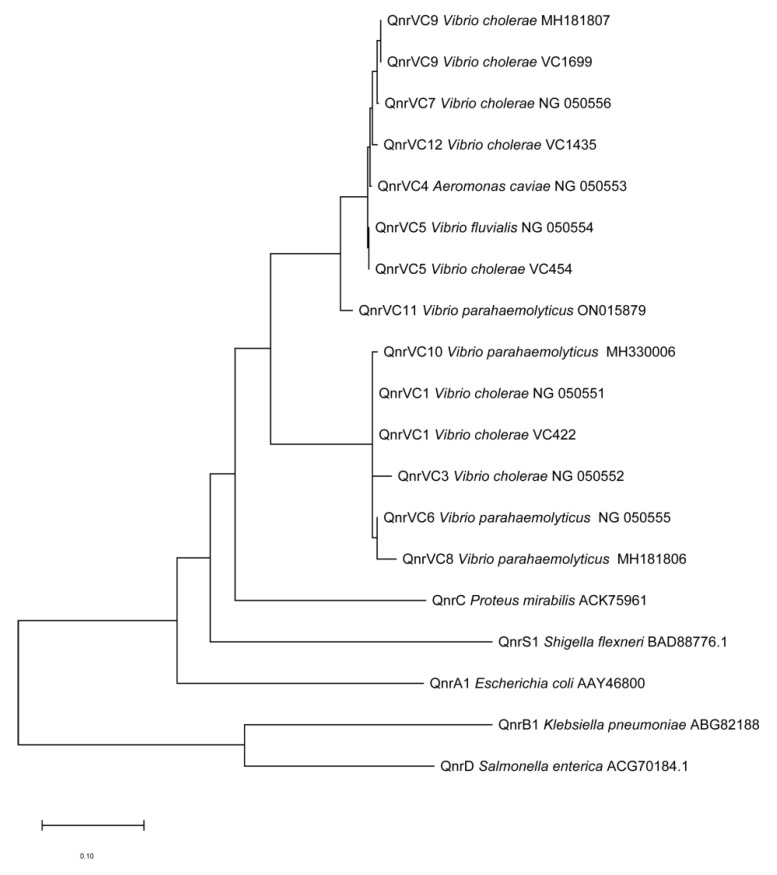
Neighbor-joining tree of Qnr protein sequences. Phylogenetic analysis of Qnr protein sequences was performed using Neighbor-Joining method by MEGA 7.0 after multiple alignment of the data via CLUSTAL 2.1. The sequences of other Qnr were obtained from GenBank and compared with QnrVC9 of VC1699, QnrVC12 of VC1435, Qnr5 of VC454 and Qnr1 of VC422.

**Figure 2 antibiotics-12-00416-f002:**
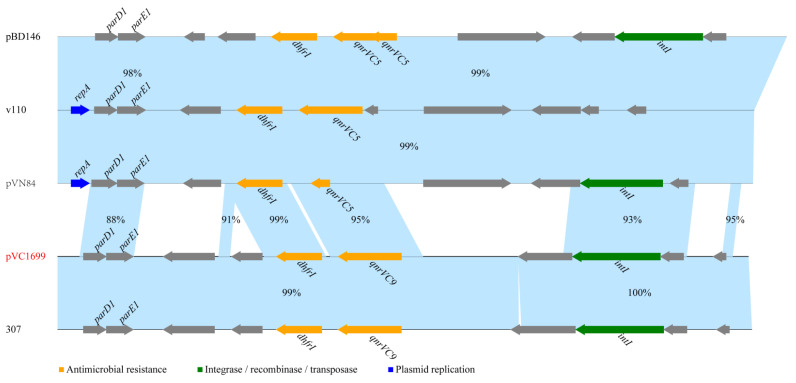
Comparing plasmid pVC1699 with sequences of four plasmids of *Vibrios* obtained from GenBank.

**Figure 3 antibiotics-12-00416-f003:**
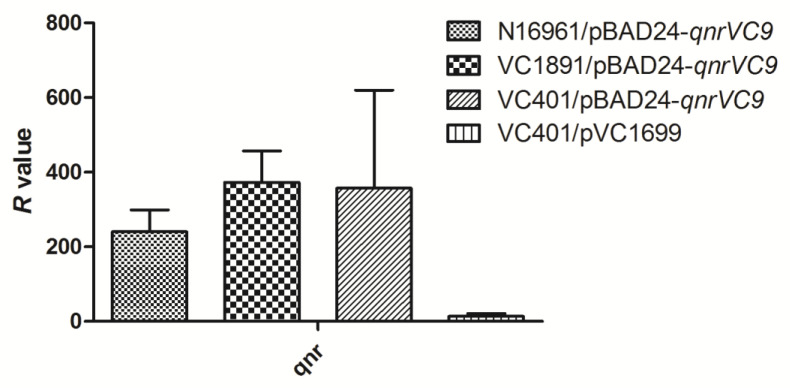
Comparisons of *qnrVC* gene expression in N16961/pBAD24-*qnrVC9,* VC1891/pBAD24-qnrVC9, VC401/pBAD24-qnrVC9, and VC401/pVC1699 cultured by Mueller–Hinton agar containing 0.1% (*w*/*v*) arabinose. The relative expression values (R) were measured by RT-PCR. All assays were performed in triplicate, and the means and standard deviations were shown.

**Table 1 antibiotics-12-00416-t001:** Characteristics of *V. cholera* O139 strains containing *qnrVC* alleles.

Strain	Year	Province	MIC (μg/mL) to CIP	*ctxAB* ^b^	Qnr	Mutation ^a^				Source
						GyrA			ParC	
						S83	D87	A171	S85	
VC422	2002	Jiangxi	0.03	-	QnrVC1	N	-	-	-	Environment ^c^
VC1435	2003	Zhejiang	0.25	+	QnrVC12	-	-	S	-	Patient
VC707	2004	Sichuan	0.03	-	QnrVC1	N	N	S	-	Environment
VC454	2004	Jiangxi	0.12	+	QnrVC5	-	-	S	-	Animal ^d^
VC319	2004	Liaoning	0.06	+	QnrVC5	-	-	S	-	Patient
VC515	2005	Sichuan	0.03	-	QnrVC1	-	-	S	-	Animal
VC1692	2006	Jiangxi	0.12	+	QnrVC5	-	-	S	-	Environment
VC1699	2006	Jiangxi	4/6	+	QnrVC9	I	-	-	L	Patient

^a^ Compared with the sequence of the *V. cholerae* O1 reference strain N16961. ^b^ *ctxAB*, Cholera toxin genes. ^c^ Surface swab of hospital, restaurant or other patient-related places, water samples from river or sewage. ^d^ Fishes, bullfrog, or soft-shelled turtle.

**Table 2 antibiotics-12-00416-t002:** Characteristics of Cloned *V. cholera* strains.

Conjugated Strains		CIP MIC (µg/mL)	QRDR Mutation		*qnr*
	0% Ara	0.1% Ara	GyrA	ParC	
VC1699	4/6	4/6	S83I	S85L	QnrVC9
VC401	0.375	0.375	S83I	S85L	-
VC401/pBAD24-*qnrVC1*	0.375	1.5	S83I	S85L	QnrVC1
VC401/pBAD24-*qnrVC5*	0.375	2	S83I	S85L	QnrVC5
VC401/pBAD24-*qnrVC12*	0.375	2	S83I	S085L	QnrVC12
VC401/pBAD24-*qnrVC9*	0.375	3/4	S83I	S85L	QnrVC9
VC401/pVC1699	4/6	4/6	S83I	S85L	QnrVC9
VC1891	0.03	0.03	A171S	-	
VC1891/pBAD24-*qnrVC9*	0.03	0.125/0.25	A171S	-	QnrVC9
N16961	0.015	0.015	-	-	-
N16961/pBAD24-*qnrVC9*	0.015	0.125	-	-	QnrVC9

**Table 3 antibiotics-12-00416-t003:** Primers used in this study.

Primer	Sequence (5′–3′)	Amplicon	Reference
*qnrAF*	ATTTCTCACGCCAGGATTTG	*qnrA*	[16]
*qnrAR*	GATCGGCAAAGGTTAGGTCA		
*qnrBF*	GATCGTGAAAGCCAGAAAGG	*qnrB*	[16]
*qnrBR2*	ATGAGCAACGATGCCTGGTA		
*qnrCF*	GGGTTGTACATTTATTGAATCG	*qnrC*	[16]
*qnrCR*	CACCTACCCATTTATTTTCA		
*qnrSmF*	GCAAGTTCATTGAACAGGGT	*qnrS*	[16]
*qnrSmR*	TCTAAACCGTCGAGTTCGGCG		
*aacIbF*	TTGCGATGCTCTATGAGTGGCTA	*aac(6′)-Ib*	[16]
*aacIbR*	CTCGAATGCCTGGCGTGTTT		
*qepAF*	AACTGCTTGAGCCCGTAGAT	*qepAF*	[16]
*qepAR*	GTCTACGCCATGGACCTCAC		
*qnrVCF*	AATTTTAAGCGCTCAAACCTCCG	*qnrVC*	[27]
*qnrVCR*	TCCTGTTGCCACGAGCATATTTT		
*qnrVC3F*	GCCATGGAAAAATCAAAGCAAT	*qnrVC3*	[27]
*qnrVC3R*	GCGGATCCGTCAGGAACAATGATTA		
*qnrF **	CGGAATTCCGAAAATGGATAAAACAGACCAGT	*qnrVC*	This study
*qnrR*	CCCAAGCTTGGGTTAGTCAGGTACTACTATTAAACCTAAT		
*2qnrF*	CGGAATTCCGATGGAAAAATCAAAGCAATTATATAATCAA	*qnrVC*	This study
*2qnrR*	CCCAAGCTTGGGTTAGTCAGGAACAATGATTACCCCTAAT		
*qnrF2*	ATTTGCCCCCTTTAAATCGCACTC	*qnrVC*	This study
*qnrR2*	TAGATTGTTCTTTCATTGAACGAG		
*recAF*	AAGATTGGTGTGATGTTTGGTA	*recA*	This study
*recAR*	CACTTCTTCGCCTTCTTTG		

* The underlined nucleotides in the primers indicate introduced restriction endonuclease recognition sites.

## Data Availability

The datasets used and/or analyzed during the current study available from the corresponding author on reasonable request.

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
