# Peer review of "Quinolone Resistance Genes and Their Contribution to Resistance in Vibrio cholerae Serogroup O139"

_antibiotics, 2023, doi:10.3390/antibiotics12020416_

Round 1
Reviewer 1 Report
Dear Authors
Thank you for the interesting study and findings. Some comments below for consideration.
Line 179: Describe how many strains from patients / the environment
Line 179: Define 'the environment' which is rather broad e.g. river, aquatic farm environment, surface swab, etc.?
Line 180: Suggest to include a table describing the breakdown details of the strains, e.g. the number of strains from sources (i.e. patients/the environment) by year for the study period (1993-2009). Also, suggest to include the trend in your analysis e.g. over a period of 1993-2009 which is more than a decade, do characteristic of V. cholerae change and if so how (to discuss).
Line 101-102: 55%, 58% identity sounds insignificant - Were they sufficient to make your conclusion? What would be the cut-off % identity the authors were looking at please?
Did the study perform MIC test for all 340 strains, if so, suggest to include these findings in the manuscript.
Did the study perform virulence identification e.g. PCR detection of specific virulence genes especially for those isolates described in Table 2? If so, suggest to incorporate the virulence genes analysis into the analysis.
Author Response
Point to Point Responses
Reviewer #1
Reviewer Comments:
Q1. Line 179: Describe how many strains from patients / the environment
A1. Thanks for the advice. We add related content: “And these strains including 237 ones that isolating from patients, 8 ones isolating from healthy carrier, 13 ones isolating from fishes, bullfrog or soft-shelled turtle, 45 ones from surface swab of patient-related places, water samples of river or sewage, and 37 ones lacking the related information.”
Q2. Line 179: Define 'the environment' which is rather broad e.g. river, aquatic farm environment, surface swab, etc.?
A2. Thanks for the advice. We add related content: “And these strains including 237 ones that isolating from patients, 8 ones isolating from healthy carrier, 13 ones isolating from fishes, bullfrog or soft-shelled turtle, 45 ones from surface swab of patient-related places, water samples of river or sewage, and 37 ones lacking the related information.” And also add related content at the table 1:
Table 1. Characteristics of V. cholera O139 strains containing qnrVC alleles
|
Strain |
Year |
Province |
MIC(μg/ml)to CIP |
ctxABb |
Qnr |
Mutation a |
Source |
||||||||
|
GyrA |
ParC |
||||||||||||||
|
S83 |
D87 |
A171 |
S85 |
||||||||||||
|
VC422 |
2002 |
Jiangxi |
0.03 0.25 0.03 0.12 0.06 0.03 0.12 8 |
- |
QnrVC1 |
N |
- |
- |
- |
Environmentc |
|||||
|
VC1435 |
2003 |
Zhejiang |
+ |
QnrVC10 |
- |
- |
S |
- |
Patient |
||||||
|
VC707 |
2004 |
Sichuan |
- |
QnrVC1 |
N |
N |
S |
- |
Environment |
||||||
|
VC454 |
2004 |
Jiangxi |
+ |
QnrVC5 |
- |
- |
S |
- |
Animald |
||||||
|
VC319 |
2004 |
Liaoning |
+ |
QnrVC5 |
- |
- |
S |
- |
Patient |
||||||
|
VC515 |
2005 |
Sichuan |
- |
QnrVC1 |
- |
- |
S |
- |
Animal |
||||||
|
VC1692 |
2006 |
Jiangxi |
+ |
QnrVC5 |
- |
- |
S |
- |
Environment |
||||||
|
VC1699 |
2006 |
Jiangxi |
+ |
QnrVC9 |
I |
- |
- |
L |
Patient |
||||||
a Compared with the sequence of the V. cholerae N16961.
b ctxAB, Cholera toxin genes.
csurface swab of hospital, restaurant or other patient-related places, water samples from river or sewage
dfishes, bullfrog or soft-shelled turtle
Q3. Line 180: Suggest to include a table describing the breakdown details of the strains, e.g. the number of strains from sources (i.e. patients/the environment) by year for the study period (1993-2009). Also, suggest to include the trend in your analysis e.g. over a period of 1993-2009 which is more than a decade, do characteristic of V. cholerae change and if so how (to discuss).
A3. Thanks for the advice. These strains were chosen using stratified sampling of cases in different years in different provinces from 1993 to 2009, and 72% of these strains were isolated from patients or healthy carrier. The described content was mainly in the related paper “ Yu L, Zhou Y, Wang R, Lou J, Zhang L, Li J, Bi Z, Kan B. Multiple antibiotic resistance of Vibrio cholerae serogroup O139 in China from 1993 to 2009. PloS one 2012, 7(6):e38633.” In this manuscript, we mainly focued on the plasmid pVC1699.
Q4. Line 101-102: 55%, 58% identity sounds insignificant - Were they sufficient to make your conclusion? What would be the cut-off % identity the authors were looking at please?
A4. Thanks for pointing out the mistake. We found 55%, 58% identity were insignificant and we have corrected the content as “BLAST revealed that pVC1699 had low identity with plasmids harbored qnrVC5, including pBD146 from V. fluvialis (GenBank accession no. EU574928.1), plasmids v110 from V. parahaemolyticus (GenBank accession no. KC540630.1) and pVN84 from V. cholerae O1 (GenBank accession no. AB200915.1) (Figure 1). But pVC1699 had high identity to 99% with a plasmid termed 307 from V. vulnificus (GenBank accession no. MZ325519) (Figure 1).”
Q5. Did the study perform MIC test for all 340 strains, if so, suggest to include these findings in the manuscript.
A5. Thanks for the advice. The study performed MIC test for all 340 strains, and show in the related paper “ Yu L, Zhou Y, Wang R, Lou J, Zhang L, Li J, Bi Z, Kan B. Multiple antibiotic resistance of Vibrio cholerae serogroup O139 in China from 1993 to 2009. PloS one 2012, 7(6):e38633.” In this manuscript, we further add related content in result part: “In a previous study, we conducted a comprehensive investigation of the antibiotic resistance of V. cholerae O139 strains isolated in China from 1993 to 2009. The multidrug resistance in O139 isolates increased suddenly and became common after 1998. Different resistance profiles were observed in the V. cholerae O139 isolates from different years, while V. cholerae O1 strains isolated in the same period were much less resistant to these antibiotics and no obvious multidrug resistance patterns were detected. We further found decreased susceptibility was exhibited to CIP in V. cholera O139 strains. Between 2001 and 2009, the median MIC of CIP was 0.5μg/ml, without a change over this interval. These values were 16.7-fold and 4.2-fold higher than the median MICs of CIP for isolates of V. cholerae in 1993-1997 (0.03μg/ml) and 1998-2000 (0.12), respectively. The seven strains resistanting to CIP, which carried these accumulated mutations in QRDRs, were isolated in 1998, 2001 and 2003, respectively. In 2006, there was one isolated strain named VC1699 possessing no mutation in QRDR of parE but resistant to CIP. ”
Q6.Did the study perform virulence identification e.g. PCR detection of specific virulence genes especially for those isolates described in Table 2? If so, suggest to incorporate the virulence genes analysis into the analysis.
A6. Thanks for the question. Table 2 showed different plasmids in the same strain VC401, which had ctxAB. ctxAB gene encoded the mainly virulence CT in V. cholera, detected by PCR screening and showed in table S1 of the paper “Multiple antibiotic resistance of Vibrio cholerae serogroup O139 in China from 1993 to 2009”. We added the related content in materials and methods part of this manuscript: “A typical wild-type V. cholerae O139 strain, VC401, harbored ctxAB gene and was chosen as the recipient strain for electroporation experiments. ” and “A total of 340 V. cholerae O139 strains recovered from patients or the environment in 1993–2009 were randomly selected from the V. cholerae strain bank at the Chinese Center for Disease Control and Prevention (Beijing, China) and were used in our previous study, including 290 toxigenic (ctxAB gene positive) strains and 50 non-toxingenic (ctxAB genes negative) strains. ”

Reviewer 2 Report
The authors studied quinolone resistance of V. cholerae O139 isolated in China and the effect of plasmid-derived factors on resistance. Plasmids are often involved in the transfer of genes between strains and species, and this manuscript suggests a link to the acquisition of quinolone resistance in Vibrio cholerae. This may be an important finding that indicates the mechanism of quinolone resistance in Vibrio cholerae.
In p. 5, L154, the authors pointed out that the qtr gene is not always on the plasmid. Please provide data showing that the qnrVC detected by PCR is PMQR in V. cholerae (Vc) O139 used by the authors. Note that the reference on p. 5, L152-154 is the reference of [15], which should be corrected.
Is there an observed increase in CIP MIC when qnrVC9 is coexisted with the wild-type reference strain (N16961)? Also, did you observe the effect of qnrVC9 in QRDR strains other than VC401 (e.g., GyrA S83N, ParC A171S)?
In Table 2, there was an increase in CIP MIC in qnrVC1, qnrVC5, and qnrVC10, respectively, but they remained low in Table 1, was this a difference in expression levels? If so, did you observe a decrease in CIP MIC in the without expression induction in Table 2? Also, what is the reason why the original plasmid (pVC1699) has a higher CIP MIC than the expression vector containing qnrVC9, it would be helpful to show the correlation with the qnrVC expression level.
Author Response
Point to Point Responses
Reviewer # 2
Reviewer Comments:
Q1.In p. 5, L154, the authors pointed out that the qnr gene is not always on the plasmid. Please provide data showing that the qnrVC detected by PCR is PMQR in V. cholerae (Vc) O139 used by the authors. Note that the reference on p. 5, L152-154 is the reference of [15], which should be corrected.
A1. Thanks for pointing out the mistake and the advice. I have corrected the reference. Except VC1699 which harbored a plasmid pVC1699, the other qnrVC-containing strains were all CIP-sensitive, so our manuscript mainly focused on pVC1699, a plasmid which could significant elevate fluoroquinolone resistance . Where the qnrVC1,5,10 located would be studied in future. We revised related content from “qnrVC4, qnrVC5, qnrVC6, and qnrVC7 resided in plasmids isolated from Vibrio species [11]. qnrVC9 resided in the V. alginolyticus chromosome [11] and we determined that Int1 captured qnrVC9 and inserted it in a plasmid.” to “qnrVC alleles were usually associated with class 1 integron, SXT or plasmid [22, 23]. Interesting, qnr genes are found in a variety of Gram-positive organisms but are chromosomal and not plasmid-mediated [9]. qnrVC9 was first reported residing in the chromosome of a V. cholera isolate obtained from a pork sample in China [19] and we determined that Int1 captured qnrVC9 and inserted it in a plasmid.” In other side, I thought this definition “plasmid-mediated quinolone resistance (PMQR) genes” might be not accurate to some extent, though qnr was first found locating on a plasmid, then it was found could be associated with class 1 integron or SXT, not only plasmid. Even qnr genes are found in a variety of Gram-positive organisms but are chromosomal and not plasmid-mediated. So we also changed the “plasmid-mediated quinolone resistance (PMQR) genes” in our manuscript to “quinolone resistance related genes”.
Q2.Is there an observed increase in CIP MIC when qnrVC9 is coexisted with the wild-type reference strain (N16961)? Also, did you observe the effect of qnrVC9 in QRDR strains other than VC401 (e.g., GyrA S83N, ParC A171S)?
A2. Thanks for the advice. We further done some experiment, and found there was an 8.3-fold increase from 0.015 to 0.125 mg/mL in CIP MIC when qnrVC9 was coexisted with the wild-type reference strain (N16961), and there was 4.2/8.3-fold increase from 0.03 to 0.125/0.25 mg/mL in CIP MIC when qnrVC9 was coexisted with VC1891, a strain had nonsense mutation GyrA A171S.
Q3.In Table 2, there was an increase in CIP MIC in qnrVC1, qnrVC5, and qnrVC10, respectively, but they remained low in Table 1, was this a difference in expression levels? If so, did you observe a decrease in CIP MIC in the without expression induction in Table 2? Also, what is the reason why the original plasmid (pVC1699) has a higher CIP MIC than the expression vector containing qnrVC9, it would be helpful to show the correlation with the qnrVC expression level.
A3. Thanks for the advice. I have repeated the antibiotic susceptibility tests and increased the dilution concentration especially the 6 mg/mL . The CIP MIC of VC401/ pBAD24-qnrVC9 was change from 3 mg/mL to 3/4 mg/mL, that of VC401/ pVC1699 was change from 8 mg/mL to 4/6 mg/mL, and there was a obvious decrease in CIP MIC of VC401/ pBAD24-qnr Without Ara expression induction. The reason that CIP MIC in VC401/pBAD24-qnrVC1, qnrVC5, qnrVC10 was lower than VC401/pBAD24qnrVC9 might be difference in sequences of qnrVC alleles. The three-dimensional structure of Qnr proteins has been determined by x-ray crystallography, essential amino acids were found in the pentapeptide repeat module and in the larger of two loops, where deletion of only a single amino acid compromises activity. In addition, RT-PCR was mainly carried out to confirm the expression of qnrVC9 gene in the native isolate VC1699 and the series of transformants. As shown in Figure 2, the mRNA expression level of qnrVC9 in pBAD24 in N16961, VC1891 and VC401 were higher than that in pVC1699 in VC401. The result illustrated that it was not the difference in expression levels coursing different CIP MIC among VC401/ pBAD24-qnr alleles. The qnrVC gene over-expression in VC401/ pBAD24-qnrVC9, its CIP MIC was still slightly lower than that of VC401/ pVC1699. This might reflect a complex interplay of many known and yet unknown genetic factors. The different strains harboring qnrVC might have different effect on CIP MIC. For example, E. coli harboring pET-qnrVC5 vector also had more than 64.2-fold increase CIP MIC from <0.000243 to 0.0156 µg/mL [12], E. coli harboring pGEMT-qnrVC1 vector had 41.7-fold increase CIP MIC from 0.003 to 0.125µg/mL [33], E. coli harboring pCR2.1–qnrVC6 vector had more than 5-fold increase CIP MIC from <0.05 to 0.25µg/mL [16]. These CIP MIC of E. coli harboring qnrVC vector was quite low though more increase in multiples than native strain. It was reported in other study that qnrVC1 seemed to have no effect when harboring by another P. aeruginosa. The CIP MIC of P. aeruginosa (pCN60qnrVC1) was only 0.19 µg/ mL, compared with >32µg/ mL for the native strain P. aeruginosa Pa25 [33].
Figure 2. Comparisons of qnrVC gene expression in N16961/ pBAD24-qnrVC9, VC1891/ pBAD24-qnrVC9, VC401/ pBAD24-qnrVC9 and VC401/ pVC1699 cultured by Mueller-Hinton agar containing 0.1% (w/v) arabinose. The relative expression values (R) were measured by RT-PCR. All assays were performed in triplicate, and the means and standard deviations are shown.
Table 2. Characteristics of Cloned V. cholera strains
|
Conjugated strains |
|
CIP MIC(µg/ml) |
QRDR mutation |
qnr |
|||
|
0% Ara |
0.1% Ara |
GyrA |
ParC |
||||
|
VC1699 |
4/6 |
4/6 |
S83I |
S85L |
QnrVC9 |
||
|
VC401 |
0.375 |
0.375 |
S83I |
S85L |
- |
||
|
VC401/ pBAD24-qnrVC1 |
0.375 |
1.5 |
S83I |
S85L |
QnrVC1 |
||
|
VC401/ pBAD24-qnrVC5 |
0.375 |
2 |
S83I |
S85L |
QnrVC5 |
||
|
VC401/ pBAD24-qnrVC10 |
0.375 |
2 |
S83I |
S085L |
QnrVC10 |
||
|
VC401/ pBAD24-qnrVC9 |
0.375 |
3/4 |
S83I |
S85L |
QnrVC9 |
||
|
VC401/ pVC1699 |
4/6 |
4/6 |
S83I |
S85L |
QnrVC9 |
||
|
VC1891 |
0.03 |
0.03 |
A171S |
- |
|
||
|
VC1891/ pBAD24-qnrVC9 |
0.03 |
0.125/0.25 |
A171S |
- |
QnrVC9 |
||
|
N16961 |
0.015 |
0.015 |
- |
- |
- |
||
|
N16961/ pBAD24-qnrVC9 |
0.015 |
0.125 |
- |
- |
QnrVC9 |
||

Round 2
Reviewer 1 Report
Thanks for considering the suggestions. OK with the responses and revised manuscript for publication.